# Synthesis and Inhibitory Studies of Phosphonic Acid Analogues of Homophenylalanine and Phenylalanine towards Alanyl Aminopeptidases

**DOI:** 10.3390/biom10091319

**Published:** 2020-09-14

**Authors:** Weronika Wanat, Michał Talma, Błażej Dziuk, Paweł Kafarski

**Affiliations:** 1Department of Bioorganic Chemistry, Wrocław University of Science and Technology, Wybrzeże Wyspiańskiego 27, 50-370 Wrocław, Poland; michal.talma@pwr.edu.pl (M.T.); pawel.kafarski@pwr.edu.pl (P.K.); 2Faculty of Chemistry, University of Opole, Oleska 48, 45-052 Opole, Poland; blazej.dziuk@pwr.edu.pl; 3Faculty of Chemistry, Wrocław University of Science and Technology, Wybrzeże Wyspiańskiego 27, 50-370 Wrocław, Poland

**Keywords:** human and porcine alanine aminopeptidase, phosphonic acid inhibitors, fluorine and bromine substitution, computer-aided simulations

## Abstract

A library of novel phosphonic acid analogues of homophenylalanine and phenylalanine, containing fluorine and bromine atoms in the phenyl ring, have been synthesized. Their inhibitory properties against two important alanine aminopeptidases, of human (hAPN, CD13) and porcine (pAPN) origin, were evaluated. Enzymatic studies and comparison with literature data indicated the higher inhibitory potential of the homophenylalanine over phenylalanine derivatives towards both enzymes. Their inhibition constants were in the submicromolar range for hAPN and the micromolar range for pAPN, with 1-amino-3-(3-fluorophenyl) propylphosphonic acid (compound **15c**) being one of the best low-molecular inhibitors of both enzymes. To the best of our knowledge, P1 homophenylalanine analogues are the most active inhibitors of the APN among phosphonic and phosphinic derivatives described in the literature. Therefore, they constitute interesting building blocks for the further design of chemically more complex inhibitors. Based on molecular modeling simulations and SAR (structure-activity relationship) analysis, the optimal architecture of enzyme-inhibitor complexes for hAPN and pAPN were determined.

## 1. Introduction

Zinc-dependent mammalian neutral aminopeptidases (APN/CD13) are considered pivotal targets for treatment diseases, characterized by their overexpression. Accordingly, APNs represent the most widely scanning transmembrane proteases belonging to M1 aminopeptidases family, with capacity to control physiological processes based on the cleavage of neutral amino acid residues from peptide and protein substrates. Dysregulation of APNs expression is revealed as a disorder of metabolic pathways leading to the development of the pathologies related mainly with tumor-cell angiogenesis and metastasis, inflammatory diseases or neuropeptides dysfunctions [1,2,3]. It is worth to mention about their well-known function as a viral receptor, including human coronaviruses, where the CD13 amino acid fragment has been defined as a key for HCoV-229E infection [4].

The potential to affect many metabolic pathways, leading to incurable diseases, prompted the search for new-targeted therapies with the manipulation of APNs activities. Except for the commercially applied natural inhibitor of APN-bestatin, numerous scientific reports describe compounds with inhibitory properties towards aminopeptidases, based on the aminophosphonate scaffold [5,6,7,8]. However, none of them were introduced to clinical studies yet and thus the design of new potent and selective molecules is desirable.

In this paper we describe series of phosphonic acid analogues of homophenylalanine and phenylalanine fluorinated and brominated in aromatic ring. We evaluated their inhibitory activities towards two aminopeptidases: human (hAPN, CD13) and porcine (pAPN). Since the sequence identity of the two enzymes is 79.3% and sequence similarity 89.4%, the porcine enzyme is often used as a model one in studies on aminopeptidase N inhibitors development. On the other hand, our previous studies [9,10] indicated that low-molecular inhibitors of these enzymes could be bound substantially differently. Therefore, we undertook studies, which are a continuation of previous works and complements the library of the phosphonic acid analogues of phenylalanine, which phenyl group is substituted with halogens. Development of the new targeted compounds is in-online with scanning low-molecular sets of compounds of similar structure as the first step to rational search for new building blocks for potential drugs. Furthermore, this method is enforced by computer-aided studies, by means molecular docking simulations. The crystallographic structures of the pAPN and hAPN have been characterized [3,11] and thus the molecular modeling was used for better understanding of structural preferences for effective inhibition.

## 2. Materials and Methods

### 2.1. Chemistry

All chemicals were purchased from commercial suppliers (Sigma Aldrich, Poznan, Poland; Trimen Chemicals, Lodz, Poland; POCh, Gliwice, Poland; ChemPur, Piekary Slaskie, Poland; Archem, Kamieniec Wroclawski, Poland, Acros Organics, Geel, Belgium, Alfa Aesar, Lancashire, United Kingdom), were of analytical grade and were used without further purification. Column chromatography was performed using silica gel 60 (70–230 mesh). Analytical thin-layer chromatography was carried out on Merck SilicaGel 60 F254 (Darmstadt, Germany) and for the visualization and detection of the compounds’ UV light at 254 nm, 2,4-dinitrophenylhydrazine and potassium permanganate were used. The ^1^H-, ^13^C-, ^19^F- and ^31^P-NMR spectra were recorded on 400YH (JEOL, Tokyo, Japan) spectrometer at 295 K operating at 400MHz (^1^H), 101MHz (^13^C), 376MHz (^19^F) and 162MHz (^31^P). Samples were diluted in chloroform-d, methanol-d4 and in the mixture of D_2_O + NaOD (99.8% at %D), with all solvents being supplied by ARMAR AG (Dottingen, Switzerland). Chemical shifts are reported relative to internal TMS (^1^H NMR), CFCl_3_ (^19^F NMR) and 85% H_3_PO_4_ (^31^P NMR) standards and are given in parts per million (ppm), while coupling constant are reported in Hz. ^19^F NMR spectra were measured without decoupling with protons. Mass spectra were recorded at the Faculty of Chemistry, Wroclaw University of Science and Technology by using a Waters LCT Premier XE mass spectrometer (electrospray ionization, ESI) (Waters, Milford, MA, USA). Melting points were determined on an SRS Melting Point Apparatus OptiMelt MPA 100 (Stanford Research Systems, Sunnyvale, CA, USA) and were reported uncorrected. All compounds were an equimolar mixture of R and S enantiomers. Analytical quality control of the final compounds was performed by HPLC (Schimadzu, Mundelein, IL, USA) and was at least above 95%. A reverse phase column (Kinetex 100A, C18, 5 µm, 150 mm × 4.6 mm) was used as combination with the following separation conditions: 99.95% H_2_O, 0.05% TFA (solvent A) and 99.95% CH_3_CN, 0.05% TFA (solvent B); 0.0–4.0 min, 0% B; 4.1–30.0 min 70% B. The flow rate was 1 mL/min and the UV signal was recorded at 220 nm and 254 nm.

### 2.2. Synthesis of Substituted Methyl Phenylpropanoates/Methyl Phenylacetates (Compounds **2** and **8**)

To the substituted phenylpropionic/phenylacetic acid (1 equiv, 0.01 mol) was added 50 mL of MeOH and the mixture was cooled to 0 °C. Then, chlorotrimethylsilane (4 equiv, 0.04 mol) was added dropwise with stirring. After the completion of addition, the reaction mixture was slowly heated to room temperature and the resulting solution was stirred at room temperature for 6–12 h. Progress of the reaction was monitored by TLC chromatography (cyclohexane/ethyl acetate 4:1, *v*/*v*). The resulting mixture was concentrated on a rotary evaporator to give the respective acid methyl ester, which was used without further purification. The structural analysis of the representative compound **2b** is showed below. The characterization data of the compounds **2c–2h** and **8a–8e** are given in Appendix A.

#### 3-(2-Fluorophenyl)propionic Acid Methyl Ester (2b)

Yellow oil, yield 100%; ^1^H NMR (400 MHz, CDCl_3_), *δ* = 7.22–7.14 (m, 2H, 2xCH_ar_), 7.02 (ddd, *J* = 18.6, 8.9, 4.8 Hz, 2H, 2xCH_ar_), 3.66 (s, 3H, OCH_3_), 2.97 (t, *J* = 7.8 Hz, 2H, CH_2_), 2.63 (t, *J* = 7.8 Hz, 2H, CH_2_) ppm; ^13^C NMR (101 MHz, CDCl_3_), *δ* = 173.22 (s, **C**OOCH_3_), 161.24 (d, *J* = 245.3 Hz, C_ar_-F), 130.67 (d, *J* = 4.8 Hz, C_ar_), 128.19 (d, *J* = 8.1 Hz, C_ar_), 127.38 (d, *J* = 15.6 Hz, C_ar_), 124.14 (d, *J* = 3.6 Hz, C_ar_), 115.37 (d, *J* = 21.9 Hz, C_ar_), 51.72 (s, COO**C**H_3_), 34.27 (d, *J* = 1.5Hz, CH_2_), 24.67 (d, *J* = 2.8 Hz, CH_2_) ppm; ^19^F NMR (376 MHz, CDCl_3_), *δ* = −118.45–−118.52 (m, 1F) ppm [12,13,14].

### 2.3. Synthesis of Substituted Phenylpropanols/Ethanols (Compounds **3** and **9**)

Sodium borohydride (7 equiv, 0.07 mol) was suspended in THF (35 mL) and the respective methyl ester in 5 mL THF was added (1 equiv, 0.01 mol). The resulting mixture was heated to reflux temperature and stirred for 15 min. Then, methanol was slowly added (15 mL) and effervescence was observed. After the completion of addition, the reaction was stirred in reflux for 20–30 min. Progress of the reaction was monitored by TLC chromatography (cyclohexane/ethyl acetate 4:1, *v*/*v*). After the reaction, solvent was removed by rotary evaporation and white paste left in the flask was dissolved in 50 mL of 1M HCl and the solution stirred overnight. The resulting liquid was neutralized by addition of 2M NaOH and then extracted with CH_2_Cl_2_ and organic layer was dried over Na_2_SO_4_. Filtration and concentration of organic residues gives the corresponding alcohol, which was used to the next step without further purification. The structural analysis of the representative compound **3b** is showed below. The characterization data of the compounds **3c–3h** and **9a–9e** are given in Appendix A.

#### 3-(2-Fluorophenyl)propanol (3**b**)

Colorless oil, yield 100%; ^1^H NMR (400 MHz, CDCl_3_), *δ* = 7.17 (tdd, *J* = 7.4, 6.4, 1.6 Hz, 2H, 2 × CH_ar_), 7.09–6.96 (m, 2H, 2xCH_ar_), 3.66 (t, *J* = 6.4 Hz, 2H, CH_2_), 2.73 (t, 2H, *J* = 7.6 Hz, CH_2_), 1.87 (ddt, *J* = 7.6, 6.4, 3.8 Hz, 2H, CH_2_), 1.53 (s, 1H, OH) ppm; ^13^C NMR (101 MHz, CDCl_3_), *δ* = 161.29 (d, *J* = 244.4 Hz, C_ar_-F), 130.77 (d, *J* = 5.1 Hz, C_ar_), 128.62 (d, *J* = 16.0 Hz, C_ar_), 127.71 (d, *J* = 8.1 Hz, C_ar_), 124.08 (d, *J* = 3.5 Hz, C_ar_)_,_ 115.30 (d, *J* = 22.3 Hz, C_ar_), 62.22 (s, CH_2_OH), 33.02 (d, *J* = 1.1 Hz, **C**H_2_-C_ar_), 25.34 (d, *J* = 2.6 Hz, CH_2_) ppm; ^19^F NMR (376 MHz, CDCl_3_), *δ* = −118.75–−118.88 (m, 1F) ppm [12,15,16].

### 2.4. Synthesis of Substituted Phenylpropionaldehydes/Phenylacetaldehydes by Using Pyridinium Chlorochromate (PCC) (Compounds **4** and **10**)

The substituted phenylpropanol or phenylethanol (1 equiv, 0.01 mol) was dissolved in 15 mL of 99.8% CH_2_Cl_2_ and 1.5 g of SiO_2_ was added. After 10 min of stirring at room temperature pyridinium chlorochromate-PCC (1.5 equiv, 0.015 mol) was added in one-portion. The mixture was intensively stirred for 30–60 min and the progress of the reaction was monitored by TLC chromatography (cyclohexane/ethyl acetate 4:1, *v*/*v*). Additionally, the aldehyde formation was observed by visualization of the TLC plate with 2,4-dinitrophenylhydrazine (2,4-DNPH) indicator. After consumption of the substrate 50 mL of diethyl ether was added and the supernatant was filtrated through a silica gel with using diethyl ether as eluent. The solvents were removed under reduced pressure and the reaction mixture (yellow oil) was purified on a silica column (cyclohexane/ethyl acetate 4:1, *v*/*v*) resulting in the desired aldehyde as colorless or yellowish oil with a specific odor. The structural analysis of the representative compound **4d** and ester **5d** (^1^H, ^13^C and ^19^F NMR spectra attached in Appendix A) are showed below. The characterization data of the compounds **4e–4h, 10a** and **10b** are given in Appendix A.

#### 2.4.1. 3-(4-Fluorophenyl)propanal (**4d**)

Colorless oil, yield 63%; ^1^H NMR (400 MHz, CDCl_3_), *δ* = 9.80 (t, *J* = 1.3 Hz, 1H, CHO), 7.15 (ddd, *J* = 8.3, 5.4, 0.5 Hz, 2H, 2xCH_ar_), 6.97 (t, *J* = 8.8 Hz, 2H, 2xCH_ar_), 2.92 (t, *J* = 7.7 Hz, 2H, CH_2_), 2.67–2.63 (m, 2H, CH_2_) ppm; ^13^C NMR (101 MHz, CDCl_3_), *δ* = 178.90 (s, CHO), 161.63 (d, *J* = 244.2 Hz, C_ar_-F), 135.82 (d, *J* = 3.3 Hz, C_ar_), 129.80 (d, *J* = 7.9 Hz, 2 × C_ar_), 115.42 (d, J = 21.2 Hz, 2 × C_ar_), 35.74 (d, *J* = 1.1 Hz, **C**H_2_-C_ar_), 29.83 (d, *J* = 0.6 Hz, CH_2_) ppm; ^19^F NMR (376 MHz, CDCl_3_), *δ* = −116.71 (tt, *J* = 8.7, 5.3 Hz, 1F) ppm [17,18].

#### 2.4.2. 3-(4-Fluorophenyl)propyl-3-(4-Fluorophenyl)propionate (**5d**)

Colorless oil, yield 25%; ^1^H NMR (400 MHz, CDCl_3_), *δ* = 7.18–7.12 (m, 2H, 2xCH_ar_), 7.11–7.05 (m, 2H, 2xCH_ar_), 6.99–6.92 (m, 4H, 4xCH_ar_), 4.06 (t, *J* = 6.5 Hz, 2H, CH**_2_**), 2.91 (t, *J* = 7.7 Hz, 2H, CH**_2_**), 2.62–2.57 (m, 4H, 2xCH_2_), 1.92–1.84 (m, 2H, CH_2_) ppm; ^13^C NMR (101 MHz, CDCl_3_), *δ* = 172.82 (s, **C**OOCH_2_), 161.50 (dd, *J* = 243.9, 14.7 Hz, 2 × C_ar_-F), 136.48 (dd, *J* = 59.9, 3.2 Hz, 2 × C_ar_), 129.79 (dd, *J* = 7.8, 2.7 Hz, 4 × C_ar_), 115.30 (dd, *J* = 21.1, 8.4 Hz, 4 × C_ar_), 63.75 (s, CH_2_), 36.01 (d, *J* = 0.9 Hz, CH_2_), 31.39 (s, CH_2_), 30.34 (s, CH_2_), 30.23 (s, CH_2_) ppm; ^19^F NMR (376 MHz, CDCl_3_), *δ* = −116.88 (tt, *J* = 8.7, 5.3 Hz, 1F), -117.31 (dq, *J* = 8.9, 5.5 Hz, 1F) ppm.

### 2.5. Synthesis of Substituted Phenylacetaldehyde/Phenylpropionaldehyde by Using Dess-Martin Periodinane (DMP) (Compounds **4** and **10**)

The substituted phenylpropanal or phenylethanal (1 equiv, 0.01 mol) dissolved in 5 mL of dry CH_2_Cl_2_ was added in one portion to the solution of Dess-Martin periodinane (1.2 equiv, 0.012 mol) in dry CH_2_Cl_2_ (20 mL). The mixture was intensively stirred at room temperature and under argon atmosphere for 10–15 min. Upon completion of the oxidation 20 mL of dichloromethane was added and the reaction mixture was extracted with a solution of saturated sodium bicarbonate. The organic phase was dried under sodium sulphate and evaporated under vacuum. The oily residue was purified by column chromatography (cyclohexane/ethyl acetate 3:1, *v*/*v*) yielding the desired aldehyde as colorless or yellowish oil with a characteristic odor. The characterization data of the compounds **4b, 4c, 10c–10e** have been presented attached in Appendix A.

### 2.6. Diphenyl 1-{[(N-Benzyloxy)carbonyl]amino}alkylphosphonates (Compounds **6** and **13**)

To the mixture of benzyl carbamate (1 equiv) and triphenyl phosphite (1 equiv) dissolved in glacial acetic acid an aldehyde (1.1 equiv) was added slowly dropwise. Then mixture refluxed for 1-2h. The solvent was removed under reduced pressure and the oily residues were dissolved in CH_3_OH and left at −20 °C for crystallization or dissolved in acetone/hexane mixture and left at 4 °C. White solid was filtered and washed with frozen methanol. When the ester did not crystallize purification by column chromatography (cyclohexane/ethyl acetate 3:1, *v*/*v*) was indispensable. The final ester was recrystallized from chloroform or dichloromethane and methanol (in ratio 1:4, *v*/*v*). The structural analysis of the representative compound **6a** is showed below. The characterization data of the compounds **6b–6h and 13a–13e** are presented in Appendix A.

Compound **6a** was prepared from commercially available aldehyde precursor.

#### Diphenyl 1-{[(*N*-benzyloxy)carbonyl]amino}-3-phenylpropylphosphonate (**6a**)

White solid, m. p. 115–116 °C; yield 60%; ^1^H NMR (400 MHz, CDCl_3_), *δ* = 7.36–7.04 (m, 20H, CH_ar_), 5.17–5.09 (m, 3H, NH + CH_2_OC), 4.58–4.46 (m, 1H, CHP), 2.87 (ddd, *J* = 14.5, 9.7, 5.1 Hz, 1H, CH_2_), 2.79–2.69 (br m, 1H, CH_2_), 2.43–2.30 (br m, 1H, CH_2_), 2.14–2.00 (br m, 1H, CH_2_) ppm; ^13^C NMR (101 MHz, CDCl_3_), *δ* = 155.95 (d, *J* = 6.0 Hz, CONH), 150.17 (dd, *J* = 23.0, 9.9 Hz, 2 × C_ar_), 140.51 (s, C_ar_), 136.13 (s, C_ar_), 129.86 (d, *J* = 10.9 Hz, 4 × C_ar_), 128.66 (s, C_ar_), 128.64 (s, 2 × C_ar_), 128.57 (s, 2 × C_ar_), 128.41 (s, 2 × C_ar_), 128.29 (s, 2 × C_ar_), 126.36 (s, C_ar_), 125.47 (d, *J* = 14.4 Hz, 2 × C_ar_), 120.68 (d, *J* = 4.1 Hz, 2 × C_ar_), 120.47 (d, *J* = 4.2 Hz, 2 × C_ar_), 67.53 (s, CH_2_Ph), 48.24 (d, *J* = 157.7 Hz, CHP), 32.06 (dd, *J* = 9.0, 4.7 Hz, CH_2_**C**H_2_CHP) ppm; ^31^P NMR (162 MHz, CDCl_3_), *δ* = 17.87 (s, 1P, *trans*), 17.57 (s, 1P, *cis*) ppm; HRMS (ESI-MS) *m/z* [MH]^+^ calculated for C_29_H_28_NO_5_P: 502.1783, found: 502.1772; [M + Na]^+^ calculated for C_29_H_28_NO_5_PNa: 524.1603, found: 524.1619 [19,20].

### 2.7. Transesterification of Diphenyl 1-{[(N-Benzyloxy)carbonyl]amino}alkylphosphonates with Methanol (Compounds **14** and **16**)

To solution of the diphenyl 1-(*N*-benzyloxycarbonyloamino)alkylphosphonate (1 equiv, 0.002 mol) dissolved in methanol (20–30 mL) anhydrous KF was added (10 equiv, 0.02 mol). To the stirred mixture the 18-crown-6 was added in catalytic amount (~20 mg). Then, the mixture was refluxed 30 min. The progress of the reaction was controlled by means of TLC (ethyl acetate/cyclohexane, 3:1, *v*/*v*). After consumption of all the substrate the solvent was evaporated under reduced pressure and the resulting solid was suspended in water and extracted with CH_2_Cl_2_ (3 × 40 mL). The organic layer was dried under sodium sulphate and evaporated under vacuum. The obtained oil was purified by column chromatography (ethyl acetate/cyclohexane, 3:1, *v*/*v*). The structural analysis of the representative compound **14b** is showed below. The characterization data of the compounds **14c, 14f, 14h**, **16d** and **16e** are given in Appendix A.

#### Dimethyl 1-{[(*N*-benzyloxy)carbonyl]amino}-3-(2-fluorophenyl)propylphosphonate (**14b**) 

White solid, m. p. 101–103 °C; yield 76%; ^1^H NMR (400 MHz, CDCl_3_), *δ* = 7.39–7.29 (m, 5H, 5xCH_ar_), 7.16 (dd, *J* = 13.7, 6.9 Hz, 2H, 2xCH_ar_), 7.06–6.94 (m, 2H, 2 × CH_ar_), 5.13 (d, *J* = 4.3 Hz, 2H, CH_2_OC, *trans*), 5.13 (d, *J* = 28.9 Hz, 2H, CH_2_OC, *cis*), 4.19–4.07 (m, 1H, CHP, *trans*), 3.71 (dd, *J* = 12.3, 10.7 Hz, 6H, 2xCH_3_), 2.89–2.81 (m, 1H, CH_2_), 2.70–2.61 (m, 1H, CH_2_), 2.20–2.09 (m, 1H, CH_2_), 1.93–1.80 (m, 1H, CH_2_) ppm; ^13^C NMR (101 MHz, CDCl_3_), *δ* = 161.17 (d, *J* = 245.0 Hz, C_ar_-F), 156.11 (d, *J* = 5.1 Hz, CONH), 136.29 (s, C_ar_), 130.84 (d, *J* = 4.8 Hz, C_ar_), 128.63 (s, 2 × C_ar_), 128.33 (s, 2 × C_ar_), 128.15 (d, *J* = 1.3 Hz, 2 × C_ar_), 124.17 (d, *J* = 3.6 Hz, C_ar_), 115.38 (d, *J* = 21.9 Hz, C_ar_), 113.18 (d, *J* = 21.0 Hz, C_ar_), 67.36 (s, **C**H_2_Ph), 53.33 (d, *J* = 7.1 Hz, OCH_3_), 53.23 (d, *J* = 6.5 Hz, OCH_3_), 47.09 (d, *J* = 156.2 Hz, **C**HP), 30.28 (d, *J* = 2.5 Hz, **C**H_2_CH_2_CHP), 25.70 (dd, *J* = 13.9, 2.4 Hz, CH_2_**C**H_2_CHP) ppm; ^19^F NMR (376 MHz, CDCl_3_), δ = −118.28–−118.37 (m, F-H, *cis*), -118.44–-118.55 (m, F-H, *trans*) ppm; ^31^P NMR (162 MHz, CDCl_3_), *δ* = 27.43 (s, 1P, *trans*), 26.94 (s, 1P, *cis*) ppm; HRMS (ESI-MS) *m/z* [MH]^+^ calculated for C_19_H_23_FNO_5_P: 396.1376, found: 396.1374; [M + Na]^+^ calculated for C_19_H_23_FNO_5_PNa: 418.1196, found: 418.1154.

### 2.8. Hydrolysis of Diphenyl and Dimethyl 1-{[(N-Benzyloxy)carbonyl]amino}alkylphosphonates (Compounds **15** and **17**)

Diphenyl or dimethyl Cbz-protected 1-aminoalkylphosphonate esters were refluxed in 10 M hydrochloric acid in acetic acid by 12–24 h (**F**) or in 12 M HCl by 6-8 h (F^*^), respectively. After evaporation of the volatile products, the obtained white solids were dissolved in EtOH and left for crystallization at room temperature. The solid products were collected by filtration and recrystallized from the hot mixture of H_2_O/EtOH. The characterization data of the final compounds **15a–15h** and **17a–17e** are showed below. The ^1^H NMR, ^1^H-^31^P HMQC and ^1^H-^13^C HMQC NMR spectra of the representative final compound **15d** were attached to Appendix A.

#### 2.8.1. 1-Amino-3-Phenylpropylphosphonic Acid (**15a**)

White solid, m. p. 296–298 °C; yield 72% from diphenyl ester; ^1^H NMR (400 MHz, D_2_O + NaOD), *δ* = 7.21–7.14 (m, 4H, 4xCH_ar_), 7.10–7.05 (m, 1H, CH_ar_), 2.71 (ddd, *J* = 13.6, 10.6, 4.9 Hz, 1H, CH_2_CH_2_CHP), 2.46 (ddd, *J* = 13.5, 10.2, 6.7 Hz, 1H, CH_2_CH_2_CHP), 2.36 (td, *J* = 10.6, 3.1 Hz, 1H, CHP), 1.93–1.81 (m, 1H, CH_2_CH_2_CHP), 1.43 (dtdd, *J* = 13.9, 10.3, 6.7, 4.9 Hz, 1H, CH_2_C**H**_2_CHP) ppm; ^31^P NMR (162 MHz, D_2_O + NaOD), *δ* = 20.87 (s, 1P) ppm; HRMS (ESI-MS) *m/z* [MH]^+^ calculated for C_9_H_14_NO_3_P: 216.0790, found: 216.0783; HPLC- retention time: 10.26 min [21,22].

#### 2.8.2. 1-Amino-3-(2-Fluorophenyl)propylphosphonic Acid (**15b**)

White solid, m. p. 292–294 °C; yield 53% from dimethyl ester; ^1^H NMR (400 MHz, D_2_O + NaOD), *δ* = 7.21 (td, *J* = 7.7, 1.3 Hz, 1H, CH_ar_), 7.10 (ddd, *J* = 13.2, 7.4, 1.6 Hz, 1H, CH_ar_), 6.97 (ddd, *J* = 18.7, 11.9, 7.9 Hz, 2H, 2 × CH_ar_), 2.82–2.72 (m, 1H, CH_2_CH_2_CHP), 2.52 (ddd, *J* = 13.8, 10.2, 6.7 Hz, CH_2_CH_2_CHP), 2.40 (td, *J* = 10.9, 2.9 Hz, CHP), 1.96–1.82 (m, 1H, CH_2_CH_2_CHP), 1.52–1.39 (m, 1H, CH_2_CH_2_CHP) ppm; ^13^C NMR (101 MHz, D_2_O + NaOD), *δ* = 160.93 (d, *J* = 242.5 Hz, C_ar_-F), 130.89 (d, *J* = 5.3 Hz, C_ar_), 129.52 (d, *J* = 16.1 Hz, C_ar_), 127.71 (d, *J* = 8.2 Hz, C_ar_), 124.30 (d, *J* = 3.4 Hz, C_ar_), 115.18 (d, *J* = 22.1 Hz, C_ar_), 50.01 (d, *J* = 138.2 Hz, CHP), 32.58 (s, CH_2_), 26.21 (dd, *J* = 13.6, 2.2 Hz, CH_2_) ppm; ^19^F NMR (376 MHz, D_2_O + NaOD), *δ* = −119.31–−119.40 (m, 1F) ppm; ^31^P NMR (162 MHz, D_2_O + NaOD), *δ* = 21.61 (s, 1P) ppm; HRMS (ESI-MS) *m/z* [MH]^+^ calculated for C_9_H_13_FNO_3_P: 234.0695, found: 234.0687; HPLC-retention time: 11.12 min.

#### 2.8.3. 1-Amino-3-(3-Fluorophenyl)propylphosphonic Acid (**15c**)

White solid, m. p. 292–294 °C; yield 55% from dimethyl ester; ^1^H NMR (400 MHz, D_2_O + NaOD), *δ* = 7.17 (td, *J* = 7.9, 6.4 Hz, 1H, CH_ar_), 6.98–6.88 (m, 2H, 2xCH_ar_), 6.80 (td, *J* = 8.8, 2.6 Hz, 1H, CH_ar_), 2.73 (ddd, *J* = 14.1, 10.7, 4.9 Hz, 1H, CH_2_CH_2_CHP), 2.49 (ddd, *J* = 13.7, 10.2, 6.6 Hz, 1H, CH_2_CH_2_CHP), 2.37 (td, *J* = 10.7, 3.1 Hz, 1H, CHP), 1.95–1.82 (m, 1H, CH_2_CH_2_CHP), 1.52–1.39 (m, 1H, CH_2_CH_2_CHP) ppm; ^13^C NMR (101 MHz, D_2_O + NaOD), *δ* = 162.70 (d, *J* = 242.6 Hz, C_ar_-F), 145.82 (d, *J* = 7.3 Hz, C_ar_), 130.05 (d, *J* = 8.5 Hz, C_ar_), 124.37 (d, *J* = 2.5 Hz, C_ar_), 115.18 (d, *J* = 20.7 Hz, C_ar_), 112.45 (d, *J* = 21.1 Hz, C_ar_), 49.79 (d, *J* = 138.1 Hz, CHP), 33.69 (s, CH_2_), 32.76 (d, *J* = 12.9 Hz, CH_2_) ppm; ^19^F NMR (376 MHz, D_2_O + NaOD), *δ* = −114.38–−114.47 (m, 1F) ppm; ^31^P NMR (162 MHz, D_2_O + NaOD), *δ* = 21.68 (s, 1P) ppm; HRMS (ESI-MS) *m/z* [MH]^+^ calculated for C_9_H_13_FNO_3_P: 234.0695, found: 234.0691; HPLC-retention time: 12.00 min.

#### 2.8.4. 1-Amino-3-(4-Fluorophenyl)propylphosphonic Acid (**15d**)

White solid, m. p. 303–305 °C; yield 86% from diphenyl ester; ^1^H NMR (400 MHz, D_2_O + NaOD), *δ* = 7.21–7.14 (m, 2H, 2xCH_ar_), 6.97–6.90 (m, 2H, 2xCH_ar_), 2.78–2.69 (m, 1H, CH_2_CH_2_CHP), 2.49 (ddd, *J* = 13.7, 10.2, 6.7 Hz, 1H, CH_2_CH_2_CHP), 2.40 (td, *J* = 10.7, 3.1 Hz, 1H, CHP), 1.95–1.84 (m, 1H, CH_2_CH_2_CHP), 1.53–1.41 (m, 1H, CH_2_CH_2_CHP) ppm; ^13^C NMR (101 MHz, D_2_O + NaOD), *δ* = 160.93 (d, *J* = 240.5 Hz, C_ar_-F), 138.82 (d, *J* = 2.8 Hz, C_ar_), 130.04 (d, *J* = 7.9 Hz, 2 × C_ar_), 115.01 (d, *J* = 21.0 Hz, 2 × C_ar_), 49.80 (d, *J* = 138.3 Hz, CHP), 34.06 (s, CH_2_), 32.17 (d, *J* = 12.9 Hz, CH_2_) ppm; ^19^F NMR (376 MHz, D_2_O + NaOD), *δ* = −118.21–−118.30 (m, 1F) ppm; ^31^P NMR (162 MHz, D_2_O + NaOD), *δ* = 21.77 (s, 1P) ppm; HRMS (ESI-MS) *m/z* [MH]^+^ calculated for C_9_H_13_FNO_3_P: 234.0695, found: 234.0689; HPLC-retention time: 11.97 min.

#### 2.8.5. 1-Amino-3-(2,4-Difluorophenyl)propylphosphonic Acid (**15e**)

White solid, m. p. 304–306 °C; yield 74% from diphenyl ester; ^1^H NMR (400 MHz, D_2_O + NaOD), *δ* = 7.19 (dd, *J* = 15.3, 8.4 Hz, 1H, CH_ar_), 6.78 (t, *J* = 8.7 Hz, 2H, 2xCH_ar_), 2.78–2.69 (m, 1H, CH_2_CH_2_CHP), 2.55–2.44 (m, 1H, CH_2_CH_2_CHP), 2.40 (td, *J* = 10.9, 3.0 Hz, 1H, CHP), 1.95–1.83 (m, 1H, CH_2_CH_2_CHP), 1.51–1.38 (m, 1H, CH_2_CH_2_CHP) ppm; ^13^C NMR (101 MHz, D_2_O + NaOD), *δ* = 160.83 (ddd, *J* = 46.8, 33.8, 11.9 Hz, 2 × C_ar_-F), 131.34 (dd, *J* = 9.5, 6.9 Hz, C_ar_), 125.30 (dd, *J* = 16.3, 3.6 Hz, C_ar_), 110.94 (dd, *J* = 20.8, 3.6 Hz, C_ar_), 103.63–103.07 (m, C_ar_), 49.94 (d, *J* = 138.2 Hz, CHP), 32.52 (s, CH_2_), 25.69 (dd, *J* = 13.6, 1.4 Hz, CH_2_) ppm; ^19^F NMR (376 MHz, D_2_O + NaOD), *δ* = −114.74–−114.84 (m, 1F), -115.23 (dd, *J* = 16.2, 8.7 Hz, 1F) ppm; ^31^P NMR (162 MHz, D_2_O + NaOD), *δ* = 21.61 (s, 1P) ppm; HRMS (ESI-MS) *m/z* [MH]^+^ calculated for C_9_H_12_F_2_NO_3_P: 252.0601, found: 252.0607; HPLC-retention time: 12.66 min.

#### 2.8.6. 1-Amino-3-(3,4-Difluorophenyl)propylphosphonic Acid (**15f**)

White solid, m. p. 292–293 °C; yield 60% from dimethyl ester; ^1^H NMR (400 MHz, D_2_O + NaOD), *δ* = 7.09–7.00 (m, 2H, 2 × CH_ar_), 6.95–6.91 (m, 2H, 2 × CH_ar_), 2.76–2.67 (m, 1H, CH_2_CH_2_CHP), 2.49 (ddd, *J* = 13.8, 10.2, 6.7 Hz, 1H, CH_2_CH_2_CHP), 2.40 (td, *J* = 10.5, 3.2 Hz, 1H, CHP), 1.97–1.84 (m, 1H, CH_2_CH_2_CHP), 1.54–1.41 (m, 1H, CH_2_CH_2_CHP) ppm; ^13^C NMR (101 MHz, D_2_O + NaOD), *δ* = 148.99 (ddd, *J* = 163.2, 151.6, 12.6 Hz, 2 × C_ar_-F), 140.12 (dd, *J* = 5.4, 3.9 Hz, C_ar_), 124.58 (dd, *J* = 6.2, 3.3 Hz, C_ar_), 116.98 (dd, *J* = 21.1, 16.7 Hz, 2 × C_ar_), 49.74 (d, *J* = 138.0 Hz, CHP), 33.66 (s, CH_2_), 32.20 (d, *J* = 12.7 Hz, CH_2_) ppm; ^19^F NMR (376 MHz, D_2_O + NaOD), *δ* = −139.66–−139.78 (m, 1F), -143.58–-143.71 (m, 1F) ppm; ^31^P NMR (162 MHz, D_2_O + NaOD), *δ* = 21.46 (s, 1P) ppm; HRMS (ESI-MS) *m/z* [MH]^+^ calculated for C_9_H_12_F_2_NO_3_P: 252.0601, found: 252.0595; HPLC-retention time: 13.58 min.

#### 2.8.7. 1-Amino-3-(4-Trifluoromethylphenyl)propylphosphonic Acid (**15g**)

White solid, m. p. 295–296 °C; yield 68% from diphenyl ester; ^1^H NMR (400 MHz, D_2_O + NaOD), *δ* = 7.41 (dd, *J* = 67.8, 8.0 Hz, 4H, 4 × CH_ar_), 2.85–2.77 (m, 1H, CH_2_CH_2_CHP), 2.58 (ddd, *J* = 13.6, 10.1, 6.7 Hz, 1H, CH_2_CH_2_CHP), 2.40 (td, *J* = 10.5, 3.2 Hz, 1H, CHP), 1.99–1.87 (m, 1H, CH_2_CH_2_CHP), 1.57–1.44 (m, 1H, CH_2_CH_2_CHP) ppm; ^13^C NMR (101 MHz, D_2_O + NaOD), *δ* = 147.56 (s, C_ar_), 129.05 (s, 2 × C_ar_), 127.27 (q, *J* = 32.0 Hz, C_ar_-CF_3_), 125.30 (q, *J* = 3.8 Hz, 2 × C_ar_), 124.53 (q, *J* = 271.0 Hz, CF_3_), 49.82 (d, *J* = 138.1 Hz, CHP), 33.70 (s, CH_2_), 32.89 (d, *J* = 12.8 Hz, CH_2_) ppm; ^19^F NMR (376 MHz, D_2_O + NaOD), *δ* = −62.32 (s, 3F) ppm; ^31^P NMR (162 MHz, D_2_O + NaOD), *δ* = 21.65 (s, 1P) ppm; HRMS (ESI-MS) *m/z* [MH]^+^ calculated for C_10_H_13_F_3_NO_3_P: 284.0663, found: 284.0668; HPLC-retention time: 16.84 min.

#### 2.8.8. 1-Amino-3-(2-Trifluoromethylphenyl)propylphosphonic Acid (**15h**)

White solid, m. p. 244–246 °C; yield 35% from dimethyl ester; ^1^H NMR (400 MHz, D_2_O + NaOD), *δ* = 7.45 (dd, *J* = 69.4, 7.7 Hz, 2H, 2 × CH_ar_), 7.31 (dt, *J* = 84.9, 7.6 Hz, 2H, 2 × CH_ar_), 2.95 (td, *J* = 13.0, 4.5 Hz, 1H, CH_2_CH_2_CHP), 2.59 (td, *J* = 12.6, 5.2 Hz, 1H, CH_2_CH_2_CHP), 2.45 (td, *J* = 10.9, 2.8 Hz, 1H, CHP), 1.98–1.85 (m, 1H, CH_2_CH_2_CHP), 1.51–1.37 (m, 1H, CH_2_CH_2_CHP) ppm; ^13^C NMR (101 MHz, D_2_O + NaOD), *δ* = 141.63 (s, C_ar_), 132.31 (s, C_ar_), 131.36 (s, C_ar_), 127.58 (q, *J* = 29.5 Hz, C_ar_-CF_3_), 126.06 (s, C_ar_), 125.88 (q, *J* = 5.8 Hz, C_ar_), 124.74 (q, *J* = 273.0 Hz, CF_3_), 50.56 (d, *J* = 137.9 Hz, CHP), 34.27 (s, CH_2_), 30.30 (d, *J* = 13.7 Hz, CH_2_) ppm; ^19^F NMR (376 MHz, D_2_O + NaOD), *δ* = −59.28 (s, 3F) ppm; ^31^P NMR (162 MHz, D_2_O + NaOD), *δ* = 21.45 (s, 1P) ppm; HRMS (ESI-MS) *m/z* [MH]^+^ calculated for C_10_H_13_F_3_NO_3_P: 284.0663, found: 284.0659; HPLC-retention time: 15.19 min.

#### 2.8.9. 1-Amino-3-(2-Bromo-4-Fluorophenyl)ethylphosphonic Acid (**17a**)

White solid, m. p. 285–287 °C; yield 56% from diphenyl ester; ^1^H NMR (400 MHz, D_2_O + NaOD), *δ* = 7.27 (dd, *J* = 8.8, 2.7 Hz, 1H, CH_ar_), 7.23 (dd, *J* = 8.6, 6.3 Hz, 1H, CH_ar_), 6.95 (td, *J* = 8.5, 2.7 Hz, 1H, CH_ar_), 3.12–3.06 (m, 1H, CH_2_CHP), 2.78 (ddd, *J* = 12.5, 10.6, 2.8 Hz, 1H, CHP), 2.62 (ddd, *J* = 14.0, 12.4, 6.5 Hz, 1H, CH_2_CHP) ppm; ^13^C NMR (101 MHz, D_2_O + NaOD), *δ* = 160.71 (d, *J* = 245.6 Hz, C_ar_-F), 136.00 (dd, *J* = 15.7, 3.4 Hz, C_ar_), 132.32 (d, *J* = 8.5 Hz, C_ar_), 124.10 (d, *J* = 9.8 Hz, C_ar_), 119.44 (d, *J* = 24.4 Hz, C_ar_), 114.44 (d, *J* = 20.8 Hz, C_ar_), 51.03 (d, *J* = 137.1 Hz, CHP), 37.04 (d, *J* = 1.6 Hz, CH_2_) ppm; ^19^F NMR (376 MHz, D_2_O+NaOD), *δ* = −115.87 (td, *J* = 8.6, 6.3 Hz, 1F) ppm; ^31^P NMR (162 MHz, D_2_O + NaOD), *δ* = 20.81 (s, 1P) ppm; HRMS (ESI-MS) *m/z* [MH]^+^ calculated for C_8_H_10_BrFNO_3_P: 297.9644, found: 297.9486; HPLC-retention time: 12.25 min.

#### 2.8.10. 1-Amino-3-(2-Bromo-5-Fluorophenyl)ethylphosphonic Acid (**17b**)

White solid, m. p. 276–277 °C; yield 30% from diphenyl ester; ^1^H NMR (400 MHz, D_2_O+NaOD), *δ* = 7.51 (dd, *J* = 8.9, 5.4 Hz, 1H, CH_ar_), 7.07 (dd, *J* = 9.5, 3.0 Hz, 1H, CH_ar_), 6.87 (td, *J* = 8.6, 3.0 Hz, 1H, CH_ar_), 3.32–3.19 (m, 2H, CH_2_CHP), 2.84 (ddd, *J* = 15.5, 12.6, 6.1 Hz, 1H, CH_2_CHP) ppm; ^13^C NMR (101 MHz, D_2_O + NaOD), *δ* = − ppm; ^19^F NMR (376 MHz, D_2_O + NaOD), *δ* = −115.15 (td, *J* = 8.8, 5.5 Hz, 1F) ppm; ^31^P NMR (162 MHz, D_2_O + NaOD), *δ* = 20.77 (s, 1P) ppm; HRMS (ESI-MS) *m/z* [MH]^+^ calculated for C_8_H_10_BrFNO_3_P: 297.9644, found: 297.9637; HPLC-retention time: 11.44 min.

#### 2.8.11. 1-Amino-3-(3-Bromo-4-Fluorophenyl)ethylphosphonic Acid (**17c**)

White solid, m. p. 289–290 °C; yield 86% from diphenyl ester; ^1^H NMR (400 MHz, D_2_O+NaOD), *δ* = 7.40 (dd, *J* = 6.8, 1.9 Hz, 1H, CH_ar_), 7.12–7.07 (m, 1H, CH_ar_), 6.98 (t, *J* = 8.7 Hz, 1H, CH_ar_), 2.98 (d, *J* = 14.1 Hz, 1H, CH_2_CHP), 2.61 (td, *J* = 11.9, 2.5 Hz, 1H, CHP), 2.27 (ddd, *J* = 14.0, 12.3, 5.9 Hz, 1H, CH_2_CHP) ppm; ^13^C NMR (101 MHz, D_2_O + NaOD), *δ* = 157.26 (d, *J* = 241.8 Hz, C_ar_-F), 138.66 (dd, *J* = 15.8, 3.6 Hz, C_ar_), 133.65 (s, C_ar_), 129.91 (d, *J* = 7.2 Hz, C_ar_), 116.07 (d, *J* = 22.0 Hz, C_ar_), 107.75 (d, *J* = 20.7 Hz, C_ar_), 52.05 (d, *J* = 137.9 Hz, CHP), 37.08 (s, *J* = 1.6 Hz, CH_2_) ppm; ^19^F NMR (376 MHz, D_2_O + NaOD), *δ* = −112.99–−113.06 (m, 1F) ppm; ^31^P NMR (162 MHz, D_2_O + NaOD), *δ* = 20.55 (s, 1P) ppm; HRMS (ESI-MS) *m/z* [MH]^+^ calculated for C_8_H_10_BrFNO_3_P: 297.9644, found: 297.9634; HPLC-retention time: 13.65 min.

#### 2.8.12. 1-Amino-3-(4-Bromo-2-Fluorophenyl)ethylphosphonic Acid (**17d**)

White solid, m. p. 280–281 °C; yield 54% from dimethyl ester; ^1^H NMR (400 MHz, D_2_O + NaOD), *δ* = 7.15 (dd, *J* = 9.8, 1.9 Hz, 2H, 2 × CH_ar_), 7.08 (t, *J* = 8.2 Hz, 1H, CH_ar_), 2.91 (d, *J* = 14.0 Hz, 1H, CH_2_CHP), 2.66–2.58 (m, 1H, CHP), 2.48–2.39 (m, 1H, CH_2_CHP) ppm; ^13^C NMR (101 MHz, D_2_O + NaOD), *δ* = 161.14 (d, *J* = 247.0 Hz, C_ar_-F), 132.80 (d, *J* = 5.7 Hz, C_ar_), 127.24 (d, *J* = 3.4 Hz, C_ar_), 126.99 (t, *J* = 15.8 Hz, C_ar_), 119.12 (d, *J* = 9.8 Hz, C_ar_), 118.52 (d, *J* = 26.0 Hz, C_ar_), 51.05 (d, *J* = 137.3 Hz, CHP), 30.87 (s, CH_2_) ppm; ^19^F NMR (376 MHz, D_2_O + NaOD), *δ* = −115.77 (t, *J* = 8.8 Hz, 1F) ppm; ^31^P NMR (162 MHz, D_2_O + NaOD), *δ* = 20.62 (s, 1P) ppm; HRMS (ESI-MS) *m/z* [MH]^+^ calculated for C_8_H_10_BrFNO_3_P: 297.9644, found: 297.9644; HPLC-retention time: 13.45 min.

#### 2.8.13. 1-Amino-3-(4-Bromo-3-Fluorophenyl)ethylphosphonic Acid (**17e**)

White solid, m. p. 291–292 °C; yield 52% from dimethyl ester; ^1^H NMR (400 MHz, D_2_O + NaOD), *δ* = 7.06 (dd, *J* = 10.1, 1.8 Hz, 1H, CH_ar_), 7.06 (dd, *J* = 10.1, 1.8 Hz, 1H, CH_ar_), 6.93 (dd, *J* = 8.2, 1.8 Hz, 1H, CH_ar_), 3.05 (ddd, *J* = 14.2, 4.9, 2.7 Hz, 1H, CH_2_CHP), 2.72 (td, *J* = 12.0, 2.7 Hz,1H, CHP), 2.37 (ddd, *J* = 14.1, 12.2, 5.9 Hz, 1H, CH_2_CHP) ppm; ^13^C NMR (101 MHz, D_2_O+NaOD), *δ* = 158.61 (d, *J* = 243.8 Hz, C_ar_-F), 143.23 (s, C_ar_), 133.10 (s, C_ar_), 126.46 (d, *J* = 3.1 Hz, C_ar_), 117.06 (d, *J* = 21.7 Hz, C_ar_), 105.34 (d, *J* = 20.8 Hz, C_ar_), 51.95 (d, *J* = 137.3 Hz, CHP), 37.36 (s, CH_2_) ppm; ^19^F NMR (376 MHz, D_2_O + NaOD), *δ* = −109.36 (dd, *J* = 10.0, 7.5 Hz, 1F) ppm; ^31^P NMR (162 MHz, D_2_O + NaOD), *δ* = 19.82 (s, 1P) ppm; HRMS (ESI-MS) *m/z* [MH]^+^ calculated for C_8_H_10_BrFNO_3_P: 297.9644, found: 297.9637; HPLC-retention time: 13.99 min.

### 2.9. Diphenyl α-Aminoalkilphosphonates (Representative Example)

#### Diphenyl 1-Amino-3-(2-Trifluoromethyl)phenylpropylphosphonate (**18h**)

White solid, m. p. 192–193 °C; yield 20%; ^1^H NMR (400 MHz, MeOD), *δ* = 7.67 (d, *J* = 7.9 Hz, 1H, CH_ar_), 7.58 (t, *J* = 7.5 Hz, 1H, CH_ar_), 7.47 (d, *J* = 7.7 Hz, 1H, CH_ar_), 7.43–7.34 (m, 5H, CH_ar_), 7.28–7.16 (m, 6H, CH_ar_), 4.27 (dt, *J* = 14.1, 6.9 Hz, 1H, CHP), 3.17 (t, *J* = 8.4 Hz, 2H, CH_2_CH_2_CHP), 2.45 (tdt, *J* = 14.0, 10.8, 6.9 Hz, 1H, CH_2_CH_2_CHP), 2.35–2.20 (m, 1H, CH_2_CH_2_CHP) ppm; ^13^C NMR (101 MHz, MeOD), *δ* = 149.53 (dd, *J* = 9.8, 5.1 Hz, 2 × C_ar_), 138.54 (s, C_ar_), 132.46 (s, C_ar_), 131.25 (s, C_ar_), 129.90 (d, *J* = 6.0 Hz, 4 × C_ar_), 128.06 (q, *J* = 29.7 Hz, C_ar_), 126.96 (s, 2xC_ar_), 126.05 (d, *J* = 5.9 Hz, 2 × C_ar_), 125.85 (q, *J* = 5.6 Hz, 2 × C_ar_), 124.67 (q, *J* = 273.0 Hz, CF_3_), 120.32 (t, *J* = 3.7 Hz, 4 × C_ar_), 46.98 (d, *J* = 138.4 Hz, CHP), 30.75 (s, CH_2_), 28.74 (d, *J* = 7.7 Hz, CH_2_) ppm; ^19^F NMR (376 MHz, MeOD), *δ* = −60.60 (s, 3F) ppm; ^31^P NMR (162 MHz, MeOD), *δ* = 13.70 (s, 1P) ppm; HRMS (ESI-MS) *m/z* [MH]^+^ calculated for C_22_H_21_F_3_NO_3_P: 436.1289, found: 436.1286.

### 2.10. Bioassay

#### 2.10.1. Enzymatic Studies

Recombinant human alanine aminopeptidase (hAPN, EC 3.4.11.2) and porcine kidney alanine aminopeptidase (pAPN, EC 3.4.11.2) were purchased as lyophilized powder from R&D System (Minneapolis, MN, Canada) and Sigma Aldrich (Poznan, Poland), respectively. Fluorogenic substrate-Ala-AMC (l-alanine-4-methylcoumaryl-7-amide) was supplied from PeptaNova (Sandhausen, Germany). Each inhibitor, enzyme and substrate were dissolved in 50 mM Tris-HCl buffer (pH = 7.00 and pH = 7.20 for hAPN and pAPN, respectively) and directly used in enzymatic screening. K_m_ values for the both enzymes towards Ala-AMC were determined analogously to the literature [23] and were 73 µM for hAPN 90 µM for pAPN. Inhibitory constants for compounds **15** and **17** were determined in 96-well plates with total volume of 100 µL at 37 °C using a spectrofluorometer (SpectraMax Gemini EM Fluorometer–Molecular Devices, San Jose, CA USA) operating at two wavelengths: excitation355 nm and emission460 nm. Before screening both of the enzymes were preincubated 30 min at 37 °C in the assay buffer before adding to the wells with appropriate concentration of the substrate and inhibitor. The fluorogenic screening measurements included: (i) eight concentrations of the respective inhibitor used in a range from 0.25 mM to 0.002 mM; (ii) 50 µM concentration of the substrate for hAPN and 100 µM for pAPN; (iii) and enzyme concentration of 0.8 nM. The monitoring of the fluorogenic group release was continued for 15 min. The linear portion of the progress curve was used to calculate velocity. For each inhibitor concentration the screening was repeated three times. The types of the inhibition and inhibitory constants were calculated by using the Lineweaver-Burk, Dixon plot and Cheng-Prusoff equation: K_i_ = IC_50_/[1 + (S/K_m_)] as described earlier [24]. Kinetics parameters were calculated using GraphPad Prism (GraphPad 7.0, GraphPad Software, San Diego, CA, USA, www.graphpad.com) and Microsoft Excel (Microsoft Corporation. (2018). *Microsoft Excel*. Retrieved from https://office.microsoft.com/excel) computer programs.

#### 2.10.2. Molecular Modelling

Crystal structures of enzymes were obtained from the Research Collaboratory for Structural Bioinformatics Protein Data Bank (RCSB-PDB): for human (*Homo sapiens*) M1 aminopeptidase 4FYT [11] and porcine (*Sus scrofa*) M1 aminopeptidase 4FKE [3]. The proteins were protonated at experimental pH. Before docking, the structures of the inhibitors and their stereochemistry were considered, protonated in experimental pH typical for each enzyme and optimized by LigPrep [25]. 

The binding modes of the inhibitors of both aminopeptidases were studied by molecular modeling. All of the compounds (both enantiomers) were docked with the use of the induced fit docking algorithm of the Maestro Schrodinger package [26]. This algorithm indicated which of the amino acids were well scored and these were therefore considered in the next step. The VSGb (variable-dielectric generalized Born) model was used, which incorporates residue-dependent effects. The solvent was water. Ligands were docked with the sample ring conformations option with a 2.5 kcal/mol energy window and standard glide, prime refinement and glide redocking (SP) procedures for the best pose for each compound [27]. MM-GBSA (molecular mechanics-generalized Born surface area) was performed as a rePrime refinement to calculate Gibbs free energies with protein flexibility, with the distances from ligands also set as 0.0 Å and 5.0 Å. The first pose with the lowest binding energy in 5.0 Å was selected as the best one.

#### 2.10.3. Crystallography

The crystals of the compounds **13a**, **13c** and **14c** were grown by slow evaporation of the chloroform solution at room temperature. The single crystals were mounted on a CCD Xcalibur diffractometer, with the graphite monochromatic, MoKα radiation (λ = 0.71073 Å) at room temperature for compounds **13a** and **14c** and at 100.0(1) K for compound **13c**. The reciprocal space was explored by ω scans with detector positions at 60 mm distance from the crystal. The diffraction data processing of studied compounds (Lorentz and polarization corrections were applied) was performed using the CrysAlis CCD [28]. All structures were solved in the monoclinic crystal system, *P*2_1_/*c* for **13a** and **14c***, P*2_1_/*n* for **13c**, space group respectively (Appendix A), by direct methods and refined by a full-matrix least-squares method using SHELXL14 program [29,30]. The H atoms were located from difference Fourier synthesis and from geometrical parameters and refined using a riding model. The structure drawings were prepared using SHELXTL and Mercury programs (Appendix A) [31].

Crystallographic data for solved structures have been deposited with the Cambridge Crystallographic Data Centre as supplementary publication number CCDC: 1968381 for **13a**, 2021711 for **13c** and 2021401 for **14c**. These data can be obtained free of charge via http://www.ccdc.cam.ac.uk/conts/retrieving.html, or from the Cambridge Crystallographic Data Centre, 12 Union Road, Cambridge CB2 1EZ, UK; fax: 144 1223 336 033; email: deposit@ccdc.cam.ac.uk.

## 3. Results and Discussion

### 3.1. Chemistry

Novel library of the phosphonic acid analogues of homophenylalanine and phenylalanine have been synthesized by multistep reaction using commercially available fluorinated 3-phenylpropionic acids and fluorinated and brominated 2-phenylacetic acids, as starting materials. Preselection of starting substrates was done basing on previous study [10] taking into consideration their availability. Because of limitation on the usefulness of the previously described multistep reaction (hydrogenolytic removal of bromine from phenyl ring during reduction of oxime to amine group) and lack of products upon synthesis of homophenylalanine analogues by this procedure, we decided to convert carboxylic acid precursors to corresponding aldehydes. Aldehydes constitute well-known class of substrates in the synthesis of the α-aminophosphonic acids by using methods based on the simultaneous formation of the P-C-N scaffold [32]. Preliminarily we evaluated amidoalkylation reaction of trivalent phosphorus compounds [33,34,35] (Scheme 1, route A). Unfortunately, we were able to isolate only ammonium chloride salt (most likely product of decomposition of reaction intermediates) and no expected products were formed. Further modifications of the amine group source (replacement of acetamide by benzyl carbamate [36,37]) also did not provide the desired product (Scheme 1, route B). Finally, the most promising method turned out to be the three-component condensation of carbonyl compound, benzyl carbamate and triphenyl phosphite, known as Birum-Oleksyszyn reaction [38,39,40].

Briefly (Scheme 2), the substituted aldehydes were obtained from aromatic carboxylic acid counterparts (compounds **1** and **7**) via formation of the methyl ester hydrochlorides (compounds **2** and **8**, route **A**) and their reduction to alcohols (compounds **3** and **9**, route **B**) [41,42]. Due to avoidance of harsh conditions, poor yields, problematic work-up upon purification we do not use direct transformation of carboxylic to hydroxyl group (e.g., by using LiAlH_4_ or LiBH_4_ and TMSCl) [43,44,45]. Although the reduction of esters to alcohols by using sodium borohydride considered as relatively difficult to achieve [46] our study confirmed relevance of this reducing agent and the pure alcohols were obtained in excellent, quantitative yields. The synthetic challenge was an oxidation of the resulting alcohols to aldehydes (compounds **4** and **10**). The procedure by Omura and Swern, where alcohols are oxidized with oxalyl chloride/DMSO, was initially examined [47]. However, as a consequence of the presence of substituents in aromatic fragment of substrates many unidentified products were obtained. Thus, we decided to evaluate Corey reagent—pyridinium chlorochromate (PCC) as an oxidizing agent (route **C**) [48,49]. Oxidation of the fluorinated 3-phenylpropylalcohols (compounds **3**) gave mixtures of two main products: the desired major product—substituted 3-phenylpropionaldehydes (compounds **4d–4h**)—and the symmetric esters (substituted 3-phenylpropyl-3-phenylpropionates - compounds **5d–5h**) as side products, most likely via oxidative esterification as reported earlier by Hunsen and Viana [50,51]. Hunsen and coworkers obtained corresponding ester upon oxidizing of phenylethanol to its suitable carbonyl derivative, by using PCC and periodic acid as co-oxidant. Viana and coworkers during the oxidation of brominated arylethers, mediated by PCC or mixture PCC/BF_3_*OEt_2_, reported the formation of esters as a main product (with 46%–76% yield). During PCC oxidation of the phenylethanols **9a** and **9b**, desired products—substituted 2-phenylacetaldehydes (compounds **10a** and **10b**) and expected esters (substituted 2-phenylacetyl-2-phenylacetates) (compounds **11a** and **11b**) were obtained. Quite unexpectedly, we also observed the formation of appropriate benzaldehydes (compounds **12a** and **12b**). Such cleavage of C-C bond during oxidation of homobenzylic alcohols by PCC was reported earlier by Fernandes [52]. The loss of the one methylene group caused by oxidative breakage of the C-C bond in the presence of PCC constitute very rare process in synthetic organic chemistry. Noteworthy, modification of molar ratio between alcohol and PCC (from 1:3 to 1:1) did not change the direction of the reaction. Moreover, elongation of the oxidation time promoted by-products production (esters or/and benzaldehydes). In order to increase the yield of the aldehydes, we decided to apply oxidation by using Dess-Martin periodinane (DMP) as catalyst (route **D**) [53,54]. The reaction was conducted in dry CH_2_Cl_2_ and the optimal molar ratio between alcohol and oxidizer was established as 1:1.2. Also the sequence of the addition of the reactants seems to play a significant role in the course of this reaction and depended on the character of the transformed alcohol. While DMP was added to the solution of alcohol, the reappearance of esters were observed, whereas the quick addition of the alcohol solution into the DMP in dry CH_2_Cl_2_ resulted only in aldehyde formation (compounds **4b**, **4c**, **10c–10e**). Cbz-protected diphenyl 1-aminoalkylphosphonates (compounds **6** and **13**), were prepared by condensation of the freshly prepared aldehyde, benzyl carbamate and triphenyl phosphite in standard conditions (route **E**) [27,55,56]. In most cases the products were precipitated from methanol at −20°C or from acetone/hexane at 4 °C (see experimental Section 2.6). Due to rotation around the C-N bond of the carbamate group in the obtained diphenyl esters they exist as mixtures of *trans*- and *cis*-forms in solutions, with *trans*-form being a predominating one [49,57]. We succeeded in obtaining two diphenyl 1-(*N*-benzyloxycarbonylamino)-2-phenylethylphosphonates (compounds **13a** and **13c**) in a form suitable for X-ray analysis (Appendix A). The final aminophosphonic acids (compounds **15a**, **15d**, **15e**, **15g**, **17a**-**17c)** were formed by acidic deprotection of the phosphoryl and amino groups in concentrated HCl in acetic acid (Scheme 3, route **F**) and further recrystallization from EtOH/H_2_O followed by adding a few drops of pyridine (to pH = 6). In the case of some diphenyl esters, the classic acidic hydrolysis (in concentrated HCl or HBr) led to removal only carbobenzoxyl group and diphenyl 1-aminoalkylphosphonates were identified (compounds **18b, 18c,** 1**8f**, **18h**, **19d** and **19e**, see experimental Section 2.9). Thus, the transesterification of Cbz-protected diphenyl 1-aminoalkylphosphonates (compounds **6b, 6c, 6f, 6h, 13d** and **13e**) with methanol in the presence of KF/18-crown-ether (Scheme 3, route **G**, see experimental Section 2.7) [58] yielded corresponding dimethyl 1-{[(*N*-benzyloxy)carbonyl]amino)alkylphosphonates **14** and **16** and their further hydrolysis with 12 M HCl (route F^*^) provided the desired products—compounds **15b**, **15c**, **15f**, **15h**, **17d**, **17e**. The crystal structure of compound **14c** was included in Appendix A).

### 3.2. Evaluation of Inhibitory Activity

Medium-throughput screening on a collection of preselected compounds against defined molecular target is one of the means for discovery of novel lead compounds or for determination of the most appropriate building blocks useful in medicinal chemistry. Thus, the obtained set of racemic phosphonic acid analogues of homophenylalanine (compounds **15**) and phenylalanine (compounds **17**) were examined for their inhibitory effects on human (hAPN) and porcine (pAPN) alanine aminopeptidases. The results of the enzymatic studies, presented in Table 1, indicate quite significant potency of these compounds towards both enzymes. The data in Table 1 are supplemented with literature inhibitory constants obtained for compounds **17t**, **17o** and **17s [10]**, which are analogues of compounds **17c**, **17d** and **17e** differing in that they contain chlorine in place of bromine in aromatic rings.

When comparing the two studied enzymes, the higher affinity of all the tested inhibitors towards hAPN over pAPNs is visible, with the differences in K_i_ values being of at least about an order in magnitude smaller for the first ones.

Phosphonic acids analogues of homophenylalanine displayed submicromolar and micromolar activity towards human and porcine aminopeptidases, respectively. Non-substituted precursor (compound **15a**) has K_i_ values, which are in good agreement with those published in the literature for its racemic mixtures (0.8 µM for hAPN and 3.69–15.9 µM for pAPN) [23,59,60]. Substitution of their phenyl moieties with fluorine atoms at any position is beneficial. Position of those substitutions seems to be less pronounced than found for analogues of phenylalanine [10]. However, it does not significantly change their affinities to both aminopeptidases, with compound **15c** being only 5 times better inhibitor than **15a**. Nonetheless, this is the most active inhibitor of hAPN amongst simple aminophosphonic acids and ranks amongst the most efficient low-molecular inhibitors of both enzymes. The substitution of the phenyl rings of phosphonic acid analogues of homophenylalanine with fluorine atoms only resulted in slightly better inhibitory potencies than compound **15a**, which indicates a good acceptance of fluorine atoms by both enzymes. It is of some importance since compound **15a** is quite commonly used as building block for the development of very potent aminopeptidase inhibitors [8].

Supplementation of the previously described set of phosphonic acid analogues of phenylalanine [10] had shown that they display lowest affinities towards the studied aminopeptidases. Albeit, the replacement of chlorine by bromine is favorable, especially in the case of porcine enzyme. Similarly to previous work [10], the substituents in *para* and *meta* positions of the phenyl ring are better accepted by binding sites of both aminopeptidases, whereas the presence of substituents in *ortho* position reduced inhibitory activities.

### 3.3. Molecular Docking

The set of compounds described in this work exhibits the best inhibitory activities determined for low molecular weight inhibitors of aminopeptidases. Therefore, it seems to be of importance to understand their interactions with enzymes at the molecular level. In order to establish architecture of enzyme-inhibitor complexes, molecular modeling techniques had been used.

Homophenylalanine analogues **15**, similarly to analogues of phenylalanine and phenylglycine studied earlier [9,10], binds in the two hydrophobic pockets S1 and S1′. Porcine aminopeptidase, despite a more open structure, has a shallower S1 pocket, consequently hPheP analogues show a higher affinity for the S1′ cavity (Figure 1).

Compound **15g**, bearing a *para*-trifluoromethyl substituent, exhibited the highest affinity to the porcine enzyme. Mode of binding of both enantiomers of this compound is also depicted in Figure 1. The obligatory prerequisite for all the aminophosphonic inhibitors are interactions of phosphonate moiety with catalytic zinc ion and tyrosine (Y472) and binding of their amino groups with acid catalytic amino acids E350 and E384. Usually stereoisomerism does not significantly affect the arrangement of the phenyl ring. It is arranged between hydrophobic amino acids from S1′ pocket and amino acids lying on the border of two pockets: G347 and A348. However, enantiomers of compound **15g** are bound differently. The *(R)*-isomer (corresponding to the *D*-isomer) is not so close to the enzyme surface, which assures interaction with asparagine (N360). Its *(S)*-enantiomer (corresponding to *L*-isomer) binds similarly but may be affected by the presence of two arginines—R358 and R376. This alongside with typical binding of the amino moiety causes, that phosphonic acid group is pushed out of the optimal position of binding and interactions with Y472 and H383. Thus, calculations predict significantly higher efficiency S isomer over R isomer.

Compared to the compound of similar activity—**15f** (Appendix A) it can be concluded that the correct stabilization of the aromatic ring and the amino group has a major influence on the affinity to pAPN.

Compounds shorter by a methylene group, analogues of phenylalanine, containing bromine in their structure have been chosen because of the large size of bromine. It would fit better to the spacious S1ꞌ site of pAPN (and most likely to the S1 site of hAPN). However, these derivatives appeared to be less active. The presence of bromine atom as a substituent negatively affects the position of the key elements of the enzyme-inhibitor puzzle. Bromine strongly interacts with H383 in the case of (*S*)-**17c**, whereas in the case of *(R)*-**17c** its lack caused that aromatic ring is somewhat pushed back in the direction of V380 (Appendix A). Thus, once more the *S* isomer is bound preferentially over *R* one. On the other hand, when the compound has bromine in the *ortho* position, as it is in the case of *(S)*-**17b**, it moves the phenyl ring away from the enzyme surface of S1ꞌ pocket (Appendix A). Quite interestingly modeling predicts that its enantiomer, compound *(R*)-**17b**, is preferentially bound in the S1 cavity. However, in this case bromine also pushes out the phenyl ring from this domain.

On the contrary to the porcine enzyme, modeling predicts that human aminopeptidase binds the aromatic fragment of the studied inhibitors in its long S1 pocket (Figure 2). Earlier studies suggested that analogues of homophenylalanine should be more efficient that analogues of phenylalanine. Indeed, elongation of the alkyl chain caused some increase in activity.

Also, in this case the most vital for the inhibitory activity is complexation of aminophosphonate fragment, which is very similar for all the inhibitors (as it is seen for representative example of compound **15c** in Figure 2). Similarly, as in case of porcine enzyme, phosphonate group is bound to zinc ion and tyrosine (Y477), whereas amine is interacting with two glutamic acids (E355 and E411). Compounds with submicromolar activity interact with the amino acids at the bottom of the S1 pocket composed of: Q211, N350; and its perpendicular wall: S895, F896, S897. These residues have the greatest influence on inhibitor binding. Aromatic ring arrangement is determined by isomerism: for *(S)*-isomers it is arranged mainly by face-to-face interactions with F896, whereas for compounds of *(R)* configuration, face-to-edge interaction is preferred. This result form more optimal binding of phosphonic moiety with participation of Y477 and form different binding of fluorine by S897.

Although modeling predicts, that **15e** and **15f** containing two fluorine substituents should increase the affinity towards the enzyme, it was not observed. The calculations predict the favorable interactions (Appendix A) for both enantiomers of compound **15f** between fluorine substituents and enzyme, by interactions with A351 and N350 observed for both isomers. These interactions simultaneously can stiffen inhibitor location at the bottom of the cavity. It is worth to mention that in the case of phenylalanine analogues, the triple substitution of phenyl ring with fluorine at 3,4 and 5 positions appeared to be highly beneficial [10]. Unfortunately, we were not able to obtain tri, tetra and penta substituted phosphonic acid analogues of the homophenylalanine.

In the case of replacement of chlorine atoms in already described inhibitors of hAPN [10] by more bulky bromine, resulting in compounds **17a–17e**, did not bring expected results and the novel inhibitors are only slightly more effective than their counterparts. Although bromine fills well the hydrophobic pocket S1 and interacts with several of its amino acid side chains simultaneous shift of the inhibitor away from the bottom of S1 cavity causes that it is not optimally bound. This is well illustrated by the architectures of the complexes of both enantiomers of compound **17b** and hAPN (Figure 3).

## 4. Conclusions

Present studies and literature data point out that the introduction of halogen atoms into the aromatic ring of phosphonic acid analogues of phenylalanine or homophenylalanine results in the elevation of the affinities of novel phosphonates towards aminopeptidases. The higher potency of homophenylalanine analogues most likely is caused by flexibility of longer aliphatic linker between aminophosphonate part of inhibitors and their phenyl rings. It ensured better fit of homophenylalanines to the hydrophobic pockets of the studied enzymes. Introduction of fluorine atoms into the phenyl ring is beneficial. Replacement of hydrogen atoms by fluorine becomes a standard since the resulting compounds are not only well accepted by enzymes and other receptor proteins but also are involved in additional interactions with them. Our study resulted in finding the most potent aminophosphonate inhibitor of hAPN-1-amino-3-(3-fluorophenyl) propylphosphonic acid (compound **15c**).

Introduction of bromine into phenyl ring of phenylalanine derivatives caused increase of inhibitory potency of novel analogues. However, these compounds are significantly less effective than analogues of homophenylalanine, which derives from bulky character of bromine resulting in lower flexibility of the inhibitor molecule. This, in turn, often resulted in its non-optimal placement in the hydrophobic binding sites of aminopeptidases.

Since porcine enzyme is considered as a good model of human one it is worth to express that aromatic aminophosphonates are bound in two, albeit similar to each other, different sites: S1 in the case of hAPN and S1′ in the case of pAPN. This explains why phosphonic acid analogues are far more effective inhibitors of human enzyme. However, the structure-activity relationship found for the two enzymes supports the possibility of using pAPN as a model of hAPN and shows that S1 and S1′ binding sites of hAPN and pAPN, respectively are quite similar. Nevertheless, some reservations is suggested here.

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
