# Peer review of "Synthesis and Inhibitory Studies of Phosphonic Acid Analogues of Homophenylalanine and Phenylalanine towards Alanyl Aminopeptidases"

_biomolecules, 2020, doi:10.3390/biom10091319_

Round 1
Reviewer 1 Report
In this manuscript a series of halogenated phosphonic analogs of phenylalanine and homophenylalanine synthesis and inhibitory studies towards human and porcine aminopeptidases is described. The introduction provides sufficient background information and the relevant references are included. The research design could be improved, by providing additional explanations as to the basis of halogen selection and the substitution pattern, perhaps indicating if this selection was based on the starting materials availability, since the pattern is not consistent between the phenylalanine and homophenylalanine analogs. According to the presented results and the conclusions, since the lengthening of the C-chain provides more effective inhibitors attributed to increased flexibility, it would be interesting to see in a future study the effect an additional methylene group would have on the inhibitory effect. The methods used are detailed, providing useful explanations to the experimental findings and the results are clearly presented. There are some minor spelling and/or grammar/syntax mistakes, for instance see lines 20-21, 39, 47, 515, 544, 571, 606Author Response
Response to Reviewer 1 Comments
Point 1. The research design could be improved, by providing additional explanations as to the basis of halogen selection and the substitution pattern, perhaps indicating if this selection was based on the starting materials availability, since the pattern is not consistent between the phenylalanine and homophenylalanine analogs.
Response 1. That preselection was done on the basis of starting materials availability. We have added a sentence about preselection for substrates, which is given in a first sentence of paragraph 3.1 (lines 423-425).
Point 2. According to the presented results and the conclusions, since the lengthening of the C-chain provides more effective inhibitors attributed to increased flexibility, it would be interesting to see in a future study the effect an additional methylene group would have on the inhibitory effect.
Response 2. The elongation of the alkyl chain by additional methylene group seems to be interesting, however there is also a possibility that it could decrease the inhibition, because of the steric perturbation (that kind of molecules could be too large for effective bindings in the APNs binding sites). However, we will take this idea into account in the future work.
Point 3. There are some minor spelling and/or grammar/syntax mistakes, for instance see lines 20-21, 39, 47, 515, 544, 571, 606
Response 3. We have followed the suggestion of the Reviewer and corrected the grammar mistakes.
Finally, we are grateful to the Referee 1 for the work done upon review of our paper and helpful comments.
Reviewer 2 Report
Wanat et al. report on the synthesis of several homophenylalanine and phenylalanine derivatives, and tested their activity towards two aminopeptidases. The findings were also supported via molecular modeling. The work reports on various new compounds and their corresponding activity. In addition, three crystal structures of small molecules are included. Overall the manuscript contains valuable information and the authors have done a large amount of work, however it lacks clarity in the explanations. Addressing these aspects will enhance the quality of the manuscript.
Several typographical errors and statements that are not clear need to be fixed. See below for some, but not all, examples (please edit carefully the text as the ones below are not all of the things needing fixes):
- The statement…to the best of our knowledge….(lines 18-21) is not clear. Homophenylalanine analogues are most active inhibitors, but then among the anionalkylphosphonic acids?? Not sure what this means.
- line 39, what is meant by commercially applied? get rid of ‘a’
- line 41, change was to were
- revise lines 43-45 sentence. Could be made clearer or split in two statements.
- line 48, what is in-online?
- line 49-50, needs to be specific on what computer-aided means
- line 50, change ‘were’ to ‘have been’. In this reviewers’ opinion, it currently reads as if the crystal structures were carried out in this work.
- line 362 – oprating
- line 376, reference missing
- line 489, the some
- line 509, get rid of ‘are’
- line 521, precoursor
- line 525, substitute word ‘anyway’
- line 528, resuled
- line 535, bindng
- line 539, replace ‘seems to be’ with ‘is’
- line 541, have? or we used?
- line 606, counterpars
Scientific merit:
There is no Section S1 supplementary materials. There is a folder called S6 and another S10 but not sure if these are connected. This reviewer could not find these files. Not sure at what stage this was lost, but it would be a lot easier if all was compiled in a single file with an index. This would facilitate a reader to look at spectra.
In this regard, a number of derivatives (2) are known, but no mention is made (ref. Leow et al. Eur. J Org. Chem. 2014, 7347), specifically 2b, 2g, 2d. what about other derivatives, are they known? This will be important to know and reference for all derivatives.
Same concern as above for compounds 3 and 9
- Supplementary materials labeled section 3, 4, or 5 are also missing. Please see above to compile and organize in a single file.
- Family of compounds 6 and 14 are new and the HRMS is appropriately reported. However this is only written for 6a and 14b. If compounds are new, the full characterization should be provided (in addition to the images).
- line 206, please clarify Z-protected. This reviewer would recommend using Cbz throughout the text, and keep it uniform.
- line 422 mentions the formation of ammonium chloride salt. It would be better to provide potential reasoning as to why it didn’t work, particularly if it has in the past (ref 22-24).
- Yields should be added to Scheme 2 and subsequent schemes
- lines 444-473 describes the oxidation process in a lot of detail. The process is interesting and thefinal protocol for the oxidation will be useful. However, in this reviewers opinion, parts of this section do not necessarily contribute to the goals of the project and could be shortened. This space could then be used elsewhere.
- lines 484-486, A statement about the molecules that were crystallized may be useful. While the crystallographic details are described within the experimental, other characteristics of these molecules may be useful.
- From scheme 3, compound 19 is missing a number 2 on the Nitrogen
- lines 486-496. This statement is not clear. It needs to be re-written, and maybe condensed, to clarify the process.
- line 511. Not sure what the superscript is describing here? how is 17ca an analogue of 17c….They are different compounds so, although analogues, they should be described with a different number for clarity.
- The abstract makes it sound as if the 3-F derivative is the most active, but then this is downplayed (as written) on the discussion (lines 525-526). It seems to this reviewer that substitution at the 3- or 4-positions is the way to go. But then, it also seems like the incorporation of F at any position is beneficial. If a 5-fold decrease is not sufficient enough, then the authors should point to what a goal would be. and what the plans are, based on these observations. Can a rational design for the next target be obtained from these data?
- In looking at Figure 1, it seems like the protein undergoes significant structural changes upon binding of the molecule, is this correct? if so it will be useful to add a comment on this. Also, it seems like only the s-isomer is interacting with Y472 and that this is one of the prerequisites (line 548). This would rule against the use of the R enantiomer, and increase the actual Ki value (if the s enantiomer is synthesized). The same is true of interactions with H383, are these meaningful?
- line 568-570 comments about the impact of the Bromine. The hypothesis for this rationalization is not clear and should be more specific. So, going back to the previous comparison, this reviewer is not clear on the impact that interactions between the small molecule and H-383 have on the overall recognition.
- The analysis of the 3-F derivative in figure 2 and in the text lacks a final conclusion on the impact that the fluorine has and how it enhances the bioactivity (or recognition in this case). What about comparison between both enantiomers? it seems, from the image, as if both have distinct ways of binding. If this is the case, which one is better?
- line 594 talks about stiffening but this needs to be clarified. Is it by the interactions between the Fluorine and other residues?
- the 3-position seems to be the most impactful. What about a meta-substituted-CF3 analogue?
- the conclusion needs to be enhanced to highlight overall trends and point other researchers to the type of derivatives that should be targeted. is stereochemistry a consideration? also, I thought that the Bromine substituent had the opposite impact from just having fluorine atoms.
Author Response
We are grateful to Reviewer 2 for evaluation of our work and their valuable comments.
Response to Reviewer 2 Comments
Point 1. The statement…to the best of our knowledge….(lines 18-21) is not clear. Homophenylalanine analogues are most active inhibitors, but then among the anionalkylphosphonic acids?? Not sure what this means.
Response 1. “To the best of our knowledge, P1 homophenylalanine analogues are the most active inhibitors of the APN among phosphonic and phosphinic derivatives described hitherto in the literature”.
Compound 15c, among the compounds with aminophosphonic acid moiety, is one of the best inhibitor towards APNs.
Point 2. - line 39, what is meant by commercially applied? get rid of ‘a’
- line 41, change was to were
- revise lines 43-45 sentence. Could be made clearer or split in two statements.
- line 48, what is in-online?
- line 49-50, needs to be specific on what computer-aided means
- line 50, change ‘were’ to ‘have been’. In this reviewers’ opinion, it currently reads as if the crystal structures were carried out in this work.
- line 362 – oprating
- line 376, reference missing
- line 489, the some
- line 509, get rid of ‘are’
- line 521, precoursor
- line 525, substitute word ‘anyway’
- line 528, resuled
- line 535, bindng
- line 539, replace ‘seems to be’ with ‘is’
- line 541, have? or we used?
- line 606, counterpars
- line 206, please clarify Z-protected. This reviewer would recommend using Cbz throughout the text, and keep it uniform.
Response 2. We have followed the suggestion of the Reviewer 2 and corrected the mistakes.
Point 3. There is no Section S1 supplementary materials. There is a folder called S6 and another S10 but not sure if these are connected. This reviewer could not find these files. Not sure at what stage this was lost, but it would be a lot easier if all was compiled in a single file with an index. This would facilitate a reader to look at spectra.
In this regard, a number of derivatives (2) are known, but no mention is made (ref. Leow et al. Eur. J Org. Chem. 2014, 7347), specifically 2b, 2g, 2d. what about other derivatives, are they known? This will be important to know and reference for all derivatives.
Same concern as above for compounds 3 and 9
- Supplementary materials labeled section 3, 4, or 5 are also missing. Please see above to compile and organize in a single file.
- Family of compounds 6 and 14 are new and the HRMS is appropriately reported. However this is only written for 6a and 14b. If compounds are new, the full characterization should be provided (in addition to the images).
Supplementary materials labeled section 3, 4, or 5 are also missing. Please see above to compile and organize in a single file.
- Family of compounds 6 and 14 are new and the HRMS is appropriately reported. However this is only written for 6a and 14b. If compounds are new, the full characterization should be provided (in addition to the images).
Response 3. The Supplementary Material was most probably submitted as *.docx file. This file is a huge one and most probably was zipped and during that operation file had been lost and only supplementary figures remained. Unfortunately, the Reviewers could not access to the information provided. The current submission includes this file as *.pdf and we hope, that its content will answer directly to some of the questions and comments of the Reviewer. We also included the suitable references (for known compounds) in text body and Supplementary Material.
Point 4. Line 422 mentions the formation of ammonium chloride salt. It would be better to provide potential reasoning as to why it didn’t work, particularly if it has in the past (ref 22-24).
Response 4. The mechanism of this multicomponent reaction is not known. Usually it provides the desired compound more or less readily. According to our unpublished results (we have performed hundreds of reactions via this procedure), the appearance of ammonium chloride is most likely result of decomposition of some intermediates of this reaction. Its appearance is usually accompanied with lack of the desired product. Thus, we added a comment on that in paragraph 3.1. “we were able to isolate only ammonium chloride salt (most likely product of decomposition of reaction intermediates) and no expected products were formed.” (lines 432-433).
Point 5. Yields should be added to Scheme 2 and subsequent schemes. Lines 484-486, A statement about the molecules that were crystallized may be useful. While the crystallographic details are described within the experimental, other characteristics of these molecules may be useful.
Response 5. Yields for the appropriate compounds (intermediates) are included in Supplementary Material. The crystallographic data of the compounds 13a, 13c and 14c are also in SM. We resigned to give yields in Schemes because they are already overcrowded with information.
Point 6. Lines 444-473 describes the oxidation process in a lot of detail. The process is interesting and the final protocol for the oxidation will be useful. However, in this reviewers opinion, parts of this section do not necessarily contribute to the goals of the project and could be shortened. This space could then be used elsewhere.
Response 6. We have shortened this part by removal of discussion of possible mechanisms of that processes.
Point 7. From scheme 3, compound 19 is missing a number 2 on the Nitrogen
Response 7. Corrected
Point 8. Lines 486-496. This statement is not clear. It needs to be re-written, and maybe condensed, to clarify the process.
Line 511. Not sure what the superscript is describing here? how is 17ca an analogue of 17c….They are different compounds so, although analogues, they should be described with a different number for clarity.
Response 8. The mentioned lines have been corrected. We renamed the chlorinated compounds as 17 adding letters according to the names from previous (cited) paper.
Point 9. The abstract makes it sound as if the 3-F derivative is the most active, but then this is downplayed (as written) on the discussion (lines 525-526). It seems to this reviewer that substitution at the 3- or 4-positions is the way to go. But then, it also seems like the incorporation of F at any position is beneficial. If a 5-fold decrease is not sufficient enough, then the authors should point to what a goal would be. and what the plans are, based on these observations. Can a rational design for the next target be obtained from these data?
Response 9. We have changed the text by adding suitable comments and explanations in paragraph 3.2 (lines 527-540)
Point 10. In looking at Figure 1, it seems like the protein undergoes significant structural changes upon binding of the molecule, is this correct? if so it will be useful to add a comment on this. Also, it seems like only the s-isomer is interacting with Y472 and that this is one of the prerequisites (line 548). This would rule against the use of the R enantiomer, and increase the actual Ki value (if the s enantiomer is synthesized). The same is true of interactions with H383, are these meaningful?
Response 10. We are particularly thankful to the Referee for this comment. Description of binding of compound 15g (Figure 1) was changed (lines 560-568).
Point 11. Line 568-570 comments about the impact of the Bromine. The hypothesis for this rationalization is not clear and should be more specific. So, going back to the previous comparison, this reviewer is not clear on the impact that interactions between the small molecule and H-383 have on the overall recognition.
Response 11. We are also thankful for this comment. The text was also changed adequately in this case (lines 580-591).
Point 12. The analysis of the 3-F derivative in figure 2 and in the text lacks a final conclusion on the impact that the fluorine has and how it enhances the bioactivity (or recognition in this case). What about comparison between both enantiomers? it seems, from the image, as if both have distinct ways of binding. If this is the case, which one is better?
Response 12. We predicted ΔG binding energies of both enantiomers. Indeed, results are not included in the text, but R enantiomer, which is consistent with the relative configuration l of natural substrates, has lower Gibbs free energy of binding in the active center:
Renantiomer – ΔG= -53.29 kcal/mol vs Senantiomer – 36.80 kcal/mol.
Point 13. Line 594 talks about stiffening but this needs to be clarified. Is it by the interactions between the Fluorine and other residues?
Response 13. Yes, it is. We have added the small information about it (lines 607-610).
Point 14. The 3-position seems to be the most impactful. What about a meta-substituted-CF3 analogue?
Response 14. We would like to thank the Referee for that suggestion. Since we will try to expand the presented studies a little bit we will take it under consideration.
Point 15. The conclusion needs to be enhanced to highlight overall trends and point other researchers to the type of derivatives that should be targeted. is stereochemistry a consideration? also, I thought that the Bromine substituent had the opposite impact from just having fluorine atoms.
Response 15. We have enlarged Conclusions a little bit taking this into consideration.
Finally, we would warmly like to thank the Reviewer 2 for the careful and detailed opinion. The comments and questions raised were very helpful and resulted in significant improvement of the manuscript (at least we hope so).
Reviewer 3 Report
In the manuscript “Synthesis and inhibitory studies of phosphonic acid
analogues of homophenylalanine and phenylalanine towards Alanyl Aminopeptidases”, authors Wanat et al describe the synthesis and evaluation of a library of phosphonic acid analogues as potential inhibitors of the alanyl aminopeptidase enzyme. They further report the identification of a potent inhibitor, compound 15c, and provide a molecular basis for inhibition using computational modeling and SAR studies.
Overall, it is an adequately written manuscript with clear presentation style, and detailed description of methods, both chemistry and biochemical related.
There are some formatting or syntax errors throughout the manuscript:
1) use of “works” instead of “previous work” in the last para of introduction;
2) use of “phenyl function” instead of “functional group” in the same para;
3) in Results section (enzymetic studies), it should be inhibitory “constants” instead of “constans”;
4) a reference shows up as “Error not found” in results in the molecular modeling section,
5) in the Discussion section 3.3 (molecular modeling) the font changes midway
I also have a few questions:
1) why is the substrate concentration used different (50 uM versus 100 uM) for the human and porcine enzymes? It hasn’t been explained in the text.
2) What is the sequence similarity/homology between the APNs from humans and pigs? Also, what is the reason for having used the porcine enzyme at all? Why not use only the human enzyme? That is not made clear.
The manuscript would be greatly improved if these concerns are addressed sufficiently by the authors, and would be then suitable for publication.
Round 2
Reviewer 2 Report
The edits made by the authors are appreciated.
The only thing that this reviewer recommends is to have a final look for grammar or syntax errors throughout the text.